# Is type of work associated with physical activity and sedentary behaviour in women with fibromyalgia? A cross-sectional study from the al-Ándalus project

Maria José Girela-Rejón [1], Blanca Gavilán-Carrera [2], Esther Aparicio-Ortega,[2] Milkana Borges-Cosic [2], Inmaculada C García-Rodríguez,[2] Manuel Delgado-Fernández [2], Fernando Estévez-López [3]

MJG-R and BG-C contributed equally.

For numbered affiliations see end of article.

**Correspondence to**
Dr Fernando Estévez-López; fer@estevez-lopez.com

## ABSTRACT

**Objectives** To analyse the association between the type of work (productive vs reproductive work) and the levels of physical activity and sedentary behaviour in women with fibromyalgia.

**Method** This cross-sectional study involved 258 women with fibromyalgia from southern Spain. Of them, 55% performed reproductive work (unpaid, associated with caregiving and domestic roles) exclusively, while 45% had productive job (remunerated, that results in goods or services). Physical activity of light, moderate and vigorous intensity in the leisure time, at home, at work, and totally were measured through the leisure time physical activity instrument and with the physical activity at home and work instrument, respectively. Sedentary behaviour was measured by the Sedentary Behaviour Questionnaire.

**Results** After adjusting for age, fat percentage, education level and marital status, the multivariate analysis of covariance model informed the existence of significant differences between type of work groups (p<0.001). Women with productive work engaged in more light physical activity at work (mean difference =448.52 min; 95% CI 179.66 to 717.38; p=0.001), and total physical activity of light (809.72 min; 535.91 to 1085.53; p<0.001) and moderate (299.78 min; 97.31 to 502.25; p=0.004) intensity. Women with reproductive work engaged in more light physical activity at home (379.14; 175.64 to 582.64; p<0.001). Leisure time physical activity and sedentary behaviour were similar in both groups (p>0.05 for all comparisons).

**Conclusions** Women with productive work had greater levels of physical activity compared with those who only did reproductive work, except for physical activity at home. Having productive work might facilitate movement of women with fibromyalgia towards a more active lifestyle.

## Strengths and limitations of this study

► The questionnaires used to assess physical activity are able to distinguish whether the behaviour is at leisure time, at home or at work.
► A large sample of patients with fibromyalgia representative of southern Spain was included.
► The cross-sectional nature of the sample impedes determination of causal relationships.
► Self-administered questionnaires could provide misleading information in women with fibromyalgia.

work. Its main characteristics are: not to be remunerated by salary, to be mainly female work and to stay invisible.[1] Nevertheless, the economically and socially recognised work is productive work (goods and services), which refers to all those human activities that have an exchange value and lead to financial remuneration.[1 2]

The society in general transmits values and rules that shape female and male identities, so that work is differentiated by gender, men being more oriented towards productive work and women being typically mainly responsible for domestic work and family care, called reproductive work.[3–5] The majority of women imagine themselves performing activities associated with their gender and live with the aim of meeting the expectations related to their maternal role. Therefore, reproductive work has been and remains to be the major difference between genders.[6 7] The lower visibility and social recognition of reproductive work also entails less economic independence as it is unpaid.[1] Despite women's increasing participation in the work market, this means to assume that women do the so-called double working day,

## INTRODUCTION

Reproductive work refers to all those tasks whose purpose is to take care and ensure the wellness of the family and the household, and is the reason why it is also called domestic

in which they face the difficulty of combining productive and reproductive work[3 8]

Fibromyalgia is a condition that affects 2.4% of the Spanish population, being more frequent among women than men.[9] Fibromyalgia is heterogeneous[10] and is characterised by widespread musculoskeletal pain of unknown origin as well as fatigue, joint stiffness and sleep problems, among others,[11 12] resulting in negative consequences on ability to work.[13] The idea that health problems are not randomly distributed, but are influenced by gender inequality is becoming increasingly popular, and the social context of women with fibromyalgia in a society assigns them the role of a carer (either for children or the elderly). Thus, family care responsibilities and the double working day that some of these women perform (remunerated work and housework) are factors that also affect their health status.[8] For instance, those women who perform productive work experience better health than those who do not in terms of pain, fatigue and depression.[14] Among the factors that could positively influence fibromyalgia symptoms, there is evidence that higher levels of physical activity of different intensities and lower levels of sedentary behaviour are associated with better symptomatology.[15–17] Unfortunately, the difficulties when coordinating productive and reproductive work could lead to a lack of time for these women,[7] which is a potential barrier to participating in physical activity.[18] This potential relationship between work type and physical activity, which may determine health in this group of patients, is, however, not well characterised as yet.

Therefore, the aim of the present study was to analyse the association between the type of work (productive vs reproductive), physical activity intensity levels (during leisure time, at home, at work and totally) and sedentary behaviour in women with fibromyalgia.

## METHOD
### Design and participants
This study belongs to the al-Ándalus project, in which a representative sample of women with fibromyalgia from Andalusia (Spain) was examined.[19] Data from the participants were collected between 1 November 2011 and 31 December 2013. Associations from the eight Andalusian provinces were contacted and other participants were recruited through email, letter, telephone or the university website. A total of 646 individuals with fibromyalgia were interested in participating in the study. The inclusion criteria for the present study were: (1) To have been diagnosed with fibromyalgia. (2) To meet the 1990 American College of Rheumatology (ACR) criteria.[11] (3) Not suffering from any acute or terminal illness, or serious cognitive impairment. (4) To be a woman. The guidelines of the Declaration of Helsinki (modified in 2000) were followed. This study also follows the standards for reporting observational studies stated in the Strengthening the Reporting of Observational studies in Epidemiology checklist initiative (online supplementary table S1).

### Procedure and instruments
The measurements were conducted in two non-consecutive days. On the first day, the participants signed an informed consent, completed a questionnaire on sociodemographic data, underwent the mini-mental state examination[20] and were examined for tender points and body composition. Between days 1 and 2, they completed the following questionnaires at home: leisure time physical activity instrument (LTPAI), physical activity at home and work instrument (PAHWI)[21] and Sedentary Behaviour Questionnaire (SBQ).[22] On the second day, the participants returned to the laboratory and questionnaires were checked by the research group.

### Sociodemographic data
The sociodemographic data were obtained through a questionnaire that included items related to educational level, marital status and working status. Working status was classified as (1) Patients who only have reproductive work (this is, unpaid and associated with caregiving and domestic roles). (2) Patients who also have productive work (this is remunerated, resulting in good or services). Women classified in productive work could do reproductive work at the same time.

### Mini-mental state examination
The MMSE was used to detect severe cognitive impairment. Its score ranges between 0 and 30, and a score under 10 was considered as cognitive impairment. It is a short test for cognitive assessment that includes questions to evaluate five areas of cognitive function: orientation, short-term memory, attention, concentration, long-term memory and language.[20]

### Tender points
The 18 tender points were assessed according to ACR guidelines[11] for the diagnosis of fibromyalgia using a standardised pressure algometer (FPK 20; Wagner Instruments, Greenwich, Connecticut, USA). Increasing pressure was applied with the algometer on the selected area and the patients were requested to inform when they started to feel pain. The tender points were classified as positive when pain appeared at a pressure of $4\,kg/cm^2$ or lower. The same procedure was repeated twice and the mean was calculated. The total number of positive tender points was recorded for every patient. The participants needed to have a minimum of 11 positive points in order to confirm the diagnosis of fibromyalgia.

### Fat percentage
The fat percentage was measured by means of a portable bioelectrical impedance analyser with eight tactile electrodes (InBody R20, Biospace, Seoul, Korea). The participants were requested to refrain from taking a shower, performing intense physical activity or eating in the 2 hours prior to the measurement. They were asked to wear only underwear and no metal objects that could interfere in the measurement.

## Leisure time physical activity instrument

The LTPAI is an instrument that has shown satisfactory reliability among patients with fibromyalgia.[21] It comprises four items with three intensity levels: light, moderate and vigorous. The participants were requested to indicate the number of hours a week they had performed leisure physical activity during the previous 4 weeks. The scale can be simplified in the following levels: (1) 0.5 to 1.5 hours per week. (2) 2 to 4 hours per week. (3) More than 4 hours per week. When a participant selected levels 1 or 2, the middle range value was used for the total score. When level 3 was selected, she was asked to provide the number of hours. When no level was selected, the number of hours was considered to be 0. The number of hours indicated by the participants for every intensity category was added up to obtain the leisure time physical activity level during 1 week.

## Physical activity at home and work instrument

The PAHWI Questionnaire[21] comprises seven items with three categories for work performed at home (light, moderate and vigorous activity) and four categories for work performed out of home (sedentary, light, moderate and vigorous activity). The participants were asked to provide the average number of hours per week that they had spent on activities of each category in the previous 4 weeks. The hours of each category were added up to obtain the total score.

## Total physical activity

Although the LTPAI and the PAHWI were administered separately, they both belong to a more complex questionnaire that covers all physical activity performed during the day: leisure time, housework and work. The three categories (light, moderate and vigorous) of both questionnaires were summed up to obtain the total number of minutes of physical activity per week. The final result was divided by seven to get the total number of minutes of physical activity per day. This custom inhouse approach has been used previously.[23]

## Sedentary Behaviour Questionnaire

The SBQ[22] informs about the time spent on 11 sedentary behaviours (watching television, sitting while eating, lying and resting, sitting while playing computer/video games, sitting while listening to music, sitting and talking on the phone, doing paperwork or office work, sitting and reading, playing a musical instrument, doing arts and crafts, sitting and driving/travelling in a car, bus or train). The 11 items were completed separately for weekdays and weekend days. Response options were 'none', '15 min or less', '30 min', '1 hour', '2 hours', '3 hours', '4 hours', '5 hours' or '6 hours or more'. The time spent on each behaviour was converted into hours (this is 15 minutes=0.25 hours). Hours per day for each item were summed separately for weekday and weekend days for the total scores of sedentary behaviours. For the summary variables of total hours/day spent in sedentary behaviours (weekday and weekend) and total sedentary hours/week, responses higher than 24 hours/day were truncated to 24 hours/day.

## Statistical analysis

Normal distribution was assumed due to the large sample size. Before main analyses and in order to identify potential confounders, the age, fat percentage, total number of tender points, educational level (ie, unfinished/primary studies or secondary/vocational/university studies) and marital status (ie, currently married or unmarried) of participants with only reproductive work and those with productive work were compared using unpaired samples t-test or $\chi^2$ tests. Given that significant differences between groups emerged for all these variables (with the exception of total number of tender points), age, fat percentage, educational level and marital status were included as covariates in all the analyses described below. Unadjusted analyses and analyses additionally accounting for disease severity assessed by the Fibromyalgia Impact Questionnaire[24] are included as supplementary material (online supplementary tables 2 and 3).

A one-way multivariate analysis of covariance (MANCOVA) was conducted to compare the mean scores of women with fibromyalgia according to their type of work (ie, participants with only reproductive work and those with productive work) on (1) Leisure time physical activity (of light, moderate and vigorous physical intensity). (2) Physical activity at home (of light, moderate and vigorous physical intensity). (3) Physical activity at work (ie, the addition of housework and work) of light, moderate and vigorous physical intensity. (4) Total physical activity (ie, the addition of leisure time, housework and work) of light, moderate and vigorous physical intensity. (5) Sedentary behaviour during weekdays. (6) Sedentary behaviour on the weekend. MANCOVA allows dependent variables to be correlated and is more powerful than analysis of covariance for detecting group differences. Statistical significance was set at p<0.05 in all analyses. The Statistical Package for Social Sciences software (IBM SPSS, V.22.0) was used. Additionally, to illustrate our significant results, violin plots were created using R package ggplot2.

## PATIENT AND PUBLIC INVOLVEMENT

Patients were not involved in the development of the research question of this study. During the recruitment process, patient associations from the eight Andalusian provinces were approached to obtain collaboration agreements and to facilitate field access. Patient associations supported recruitment by disseminating advertisements for study participation. Individual patients were also contacted and informed about the study aims by the research team through email, letter, telephone or the university website. During the conduct of the study, patients' previous experiences in the initial phases of the al-Ándalus project were considered to create

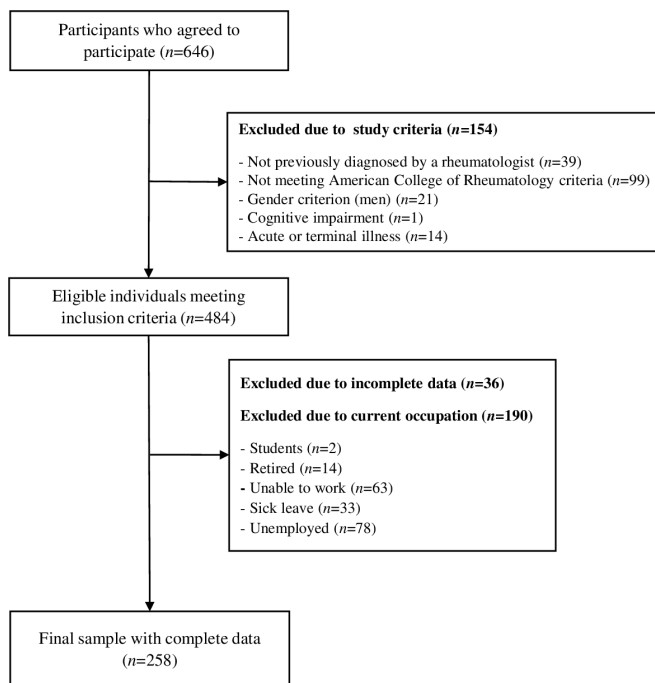

**Figure 1** Flow diagram for study participants.

the assessment protocol. The assessment of different outcomes was planned so patients have enough time to fill questionnaires, rest and recover after exhausting tests. Several talks have been carried out in the associations, at conferences and in media to disseminate the results of the al-Ándalus project.

## RESULTS

The flow chart of the participants included in the present study is shown in figure 1. The final sample was composed of 258 women with fibromyalgia. Table 1 shows the sample characteristics. The majority of the participants was married (77.9%) and had low educational level (incomplete studies 9.7%; primary school 51.2%; secondary school 25.2%). Of the women 55.4% performed reproductive work, while 44.6% performed productive work. The participants' mean age was 51 years and their mean fat percentage was 40.2%. Statistically significant differences emerged between groups (productive vs reproductive) with regard to age, body fat percentage, educational level and marital status (all, P values ≤0.001) but not with regard to the total number of tender points (data not shown but available on request).

After adjusting for age, fat percentage, educational level and marital status, the MANCOVA model showed the existence of significant differences between type of work groups; V=0.605, F (12, 241)=30.763, p<0.001. In particular comparisons, we found no differences in leisure time physical activity between work groups at any intensity level (all, p>0.05). Comparisons for physical activity at home revealed that those women with reproductive work had greater levels of light physical activity at home (mean difference=379.14, 95 % CI 175.64 to 582.64,

**Table 1** Sociodemographic characteristics of the participants (n=258)

| | Frequency | % |
|---|---|---|
| **Educational level** | | |
| Incomplete studies | 25 | (9.7) |
| Primary school | 132 | (51.2) |
| Secondary school | 65 | (25.2) |
| University degree | 36 | (14.0) |
| **Marital status** | | |
| Married | 201 | (77.9) |
| Single | 14 | (5.4) |
| Separated/divorced | 26 | (10.1) |
| Widowed | 17 | (6.6) |
| **Current working status** | | |
| Reproductive work | 143 | (55.4) |
| Productive work | 115 | (44.6) |
| | **Mean** | **(SD)** |
| Age | 51.4 | (7.9) |
| Fat percentage (%) | 40.2 | (7.3) |
| Total number of tender points | 16.6 | (2.0) |
| **Leisure-time PA (LTPAI, min/week)** | | |
| Light (<3 METs) | 178.9 | (143.0) |
| Moderate (3–6 METs) | 62.8 | (100.3) |
| Vigorous (>6 METs) | 16.5 | (43.6) |
| **PA at home (PAHWI, min/week)** | | |
| Light (<3 METs) | 1188.8 | (786.2) |
| Moderate (3–6 METs) | 544.4 | (521.0) |
| Vigorous (>6 METs) | 70.4 | (249.2) |
| **PA at work (reproductive and productive, min/week)** | | |
| Light | 1542.5 | (1003.2) |
| Moderate | 607.2 | (553.1) |
| Vigorous | 86.9 | (258.8) |
| **Total sedentary time (SBQ)** | | |
| Monday to Friday (min/weekday) | 533.1 | (261.2) |
| During the weekend (min/weekend day) | 566.2 | (257.6) |

LTPAI, leisure time physical activity instrument ; METs, metabolic index measurement unit; PA, physical activity; PAHWI, physical activity at home and work instrument; SBQ, sedentary behaviour questionnaire.

p<0.001, figure 2) and that there were no statistically significant differences for the remaining comparisons of moderate (p=0.081) or vigorous (p=0.101) physical activity. Comparisons for physical activity at work (ie, house + work) revealed that those women with productive work had greater levels of light physical activity at work (mean difference=448.52, 95 % CI 179.66 to 717.38, p=0.001, figure 3) and that there were no statistically significant differences for the remaining comparisons

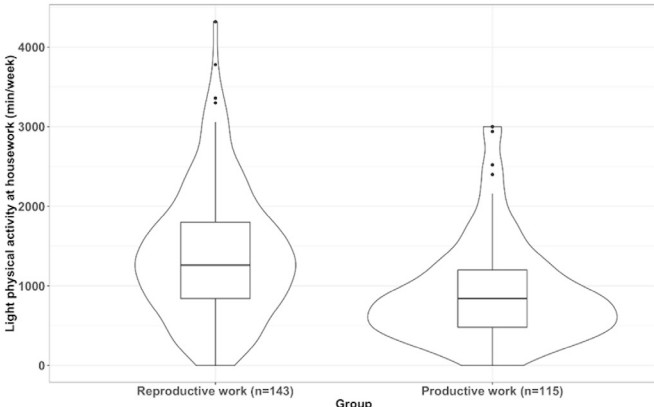

**Figure 2** Differences in physical activity at home of light intensity between women with fibromyalgia who only performed reproductive work and those who also performed productive work (p<0.001).

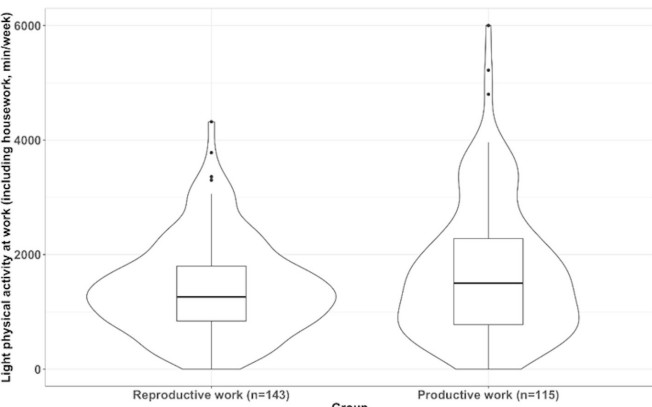

**Figure 3** Differences in physical activity at work (ie, house + work) of light intensity between women with fibromyalgia who only performed reproductive work and those who also performed productive work (p=0.001).

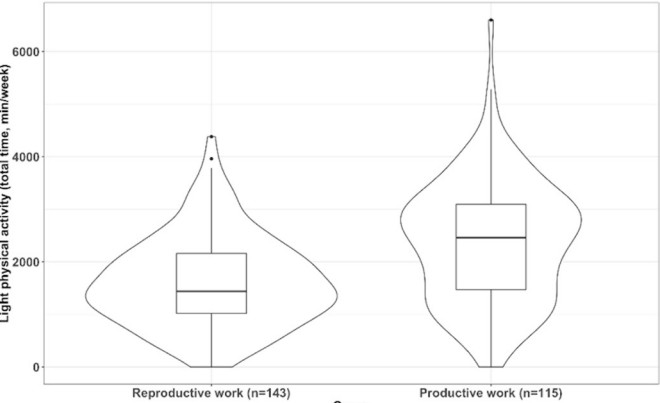

**Figure 4** Differences in total physical activity (ie, the addition of leisure time, house and work) of light intensity between women with fibromyalgia who only performed reproductive work and those who also performed productive work (p<0.001).

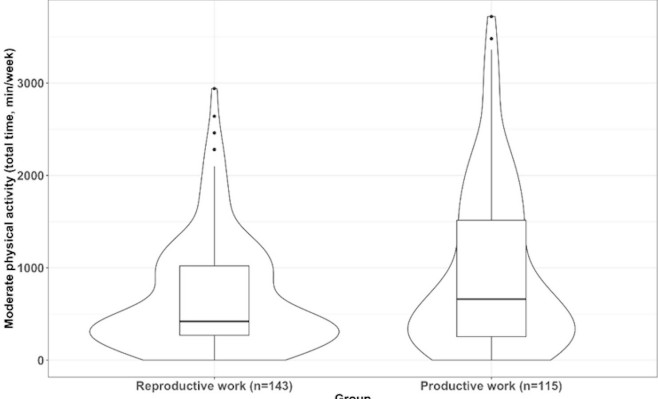

**Figure 5** Differences in total physical activity (ie, the addition of leisure time, house and work) of moderate intensity between women with fibromyalgia who only performed reproductive work and those who also performed productive work (p=0.004).

of moderate (p=0.142) or vigorous (p=0.920) physical activity. Comparisons for total physical activity (ie, the addition of leisure time, house and work) revealed that those women with productive work had greater levels of light (mean difference=809.72, 95 % CI 535.91 to 1085.53, p<0.001, figure 4) and moderate physical activity (mean difference=299.78, 95 % CI 97.31 to 502.25, p=0.004, figure 5) but not of vigorous physical activity (p=0.089). No differences for sedentary behaviour either on weekdays (p=0.308) or weekends were found (p=0.749). Sensitivity analyses additionally considering disease severity did not alter the results (online supplementary tables 2 and 3).

## DISCUSSION

In the present study, a number of differences were detected with regard to physical activity between women with fibromyalgia according to their type of work. In particular, women with only reproductive work did more light intensity physical activity at home. However, women with productive work engage more often in total physical activity (ie, the addition of physical activity at leisure, home and work) of light and moderate intensity. The remaining comparisons of physical activity in leisure time and sedentary behaviour during weekdays and weekends, yielded no statistically significant differences between both groups.

Our results revealed that both the group of women who did reproductive work and the group of women who also performed productive work obtained similar results in light, moderate and vigorous physical activity during their leisure time. According to a survey on sport habits in Spain,[18] lack of time is the major barrier to sport practice. Women, irrespective of whether they work out of home or not, continue to be mostly responsible for domestic work,[3] which suggests that this work takes most of their leisure time. Moreover, according to Teniente,[6] most women visualise themselves performing activities traditionally associated with their gender. Also, female sport is still in a situation of inequality, where sport is considered a social activity related to men because they are the

only ones supposed to have the physical characteristics to do it.[6] Some authors support the fact that the main barriers to physical activity for women are family care responsibilities[25 26] and leisure time physical activity is reduced in women with family responsibilities.[27] In this regard, reproductive work depends on the family needs and requires dedication with no fixed schedule, turning leisure time into residual and variable time. Furthermore, those women who do productive work and also take care of the bulk of unpaid work are in fact completing a double working day, so that their free time is reduced too.[28] Therefore, we hypothesised that no differences were observed between the two groups as both groups performed little physical activity due to lack of time because of family care or work responsibilities.

Light physical activity is the most common physical activity intensity in women with fibromyalgia.[29] In the present study, the group of women with fibromyalgia who did only reproductive work showed higher levels of light physical activity (but not moderate or vigorous physical activity) at home. Spitze and Loscocco[30] pointed out that women who exclusively performed reproductive work spent more hours on housework because they had more time and they did not have to bear what is known as double day. This could be the reason why higher light physical activity during domestic work was observed among women who only did reproductive work, compared with the group who also did productive work. With regard to work, however, we found that women with productive work engage more often in light physical activity at work; as the sum of physical activity at both types of (reproductive and productive) work. These findings could be explained by the so-called double day, this is, women with reproductive work often perform most of the housework at their homes in addition to their work hours.[31–33] Indeed, reproductive work has been traditionally assigned to women and this is as a result of gender differences,[3 4 34] no matter whether women have incorporated the productive sphere or not.[34] Furthermore, women who perform productive work are subject to their job demands, which prevent them from decreasing their performance or taking long breaks during their working hours.[33] Thus, also in agreement with the findings from Álvarez-Gallardo et al,[35] women who did productive work reported higher total physical activity levels compared with those who exclusively did reproductive work. Active commuting might be another potential cause of the increase in physical activity in the group of women who did productive work,[36] although other studies failed to find an association between work status and physical activity.[27]

In the present study, women with fibromyalgia who only did reproductive work and those who additionally did productive work engaged for a similar amount of time in sedentary behaviour both during weekdays and weekends. Although previous studies have revealed that physically demanding jobs are associated with increased pain and fatigue,[37] the appropriate approach for these women would not be to make them more sedentary but, as stated by Galiano and Sañudo,[38] to let them perform physical activity at an intensity that leads to positive effects, but not as high as to increase the symptoms. In this regard, physical activity improves aspects such as quality of life of women with fibromyalgia,[39] preventing problems derived from a sedentary lifestyle. Therefore, we believe that an adaptation consisting of doing more sedentary work is not advisable for women with fibromyalgia. In contrast, it would be ideal to make individual adaptations for every woman and her characteristics. However, work adaptations for people with fibromyalgia are not very frequent.[40]

Although the cross-sectional design of the present study precludes suggesting the potential clinical implications of our findings, speculation might be done accordingly. As suggested by Kivimäki et al,[41] inability to work is often a consequence of fibromyalgia in women with this condition, so it would be recommendable to make adaptations of their working conditions. In relation to this, Henriksson, Liedberg and Gerdle[42] stated that adaptations made at work are key determinants of job maintenance. This is why these authors recommend establishing adaptation strategies such as modifying the tasks, reducing the working hours, determining their own schedule based on their state, working from home or adapting at the working place. Therefore, in future research it would be advisable to go deeper into work adaptations and to assess the need for greater personalisation. Interestingly, Palstam et al[13] confirmed that people with fibromyalgia who performed productive work enjoyed better health than those not doing such work, especially in terms of pain, fatigue, stiffness, depression, physical aspects of quality of life and general state of health. For this reason, work was considered a relevant factor in the state of health of women with fibromyalgia,[14] suggesting that they can continue working with no negative consequences for their condition as long as required adaptations are taken. It is important to bear in mind that to have productive job is considered one of the main activities in one's life, so adopting strategies to stay active would contribute to personal development and self-esteem in women with fibromyalgia.[31]

The conclusions of the present study must be taken bearing its limitations in mind. A cross-sectional design was used, so the results cannot be deemed as causal. The results cannot be extended to the rest of the Spanish population since this study has been conducted in Andalusia. Furthermore, self-administered questionnaires are often seen as providers of misleading information in women with fibromyalgia.[43] However, objective tools such as accelerometers are not able to distinguish between whether the behaviour is being done at leisure time, at home or at work. Thus, further research using objective data of physical activity and sedentary behaviour at different settings is warranted when new engineering and data science developments allow it.[44 45] While these developments are not performed, subjective and objective information may be better seen as complementary.[23 46] Other limitation is that type of activity (eg, mostly either sitting, standing, walking or physical labour) performed

during work was not considered in the present study. For instance, previous studies have observed that different type of jobs (ie, mostly either sitting, standing, walking or physical labour) are concurrently associated with levels of physical activity and sedentary behaviour.[47 48] Thus, future research should account for type of activity during (productive and reproductive) work to comprehensively understand behavioural patterns of people with fibromyalgia at work.

To conclude, our study revealed that, in general, differences were found in work and home-related (but not leisure time) physical activity between women with fibromyalgia who exclusively performed reproductive work and those who also did productive work. In particular, women with only reproductive work did more light physical activity at home. However, women with productive work engage more often in light physical activity at work (ie, work + home) and total physical activity (ie, the addition of physical activity at leisure, home and work) of light and moderate intensity. The remaining comparisons did not yield statistically significant differences between both groups, indicating similar physical activity in leisure time and sedentary behaviour during weekdays and weekends. Given the cross-sectional design of this study, future longitudinal research is warranted in order to address the causality of the present findings. If they are corroborated, a robust suggestion would be that to have productive work might promote higher physical activity levels in women with fibromyalgia. Thus, a next step in research would be to find adaptations at work that help balance engagement in bouts of physical activity of adequate length and intensity tailored to women with fibromyalgia capabilities.

**Author affiliations**
[1]Physical Activity for HEaLth Promotion research group (PA-HELP), Sport and Health University Research Institute (iMUDS), Department of Didactic of Corporal Expression, Faculty of Education Sciences, University of Granada, Granada, Spain
[2]Physical Activity for HEaLth Promotion research group (PA-HELP), Sport and Health University Research Institute (iMUDS), Department of Physical Education and Sports, University of Granada, Granada, Spain
[3]Department of Child and Adolescent Psychiatry/Psychology, Erasmus MC University Medical Center, Rotterdam, The Netherlands

**Contributors** MJG-R, MD-F, FE-L: Conception or design of the work, acquisition of data, analysis and interpretation of data, and drafting the work for intellectual content; BG-C, EA-O, MB-C, ICG-R: Analysis and interpretation of data and drafting the work for intellectual content.

**Funding** This work was supported by the Spanish Ministries of Economy and Competitiveness (I+D+i DEP2010-15639; I+D+i DEP2013-40908-R) and the Spanish Ministry of Education (FPU15/00002). This study has been partially funded by the University of Granada, Plan Propio de Investigación 2016, Excellence actions: Units of Excellence; Unit of Excellence on Exercise and Health (UCEES), and by the Junta de Andalucía, Consejería de Conocimiento, Investigación y Universidades and European Regional Development Fund (ERDF), ref. SOMM17/6107/UGR. FE-L has received funding from the European Union's Horizon 2020 research and innovation programme under the Marie Skłodowska-Curie grant agreement no. 707404. The funders of this study did not have any role in the study design, data collection and analyses, decision to publish or preparation of the manuscript.

**Competing interests** None declared.

**Patient and public involvement** Patients and/or the public were not involved in the design, or conduct, or reporting, or dissemination plans of this research.

**Patient consent for publication** Not required.

**Ethics approval** The al-Ándalus project protocol was approved by the ethics committee of Virgen de las Nieves Hospital (Granada, Spain) with register number: 15/11/2013 N72.

**Provenance and peer review** Not commissioned; externally peer reviewed.

**Data availability statement** Data are available upon reasonable request.

**ORCID iDs**
Maria José Girela-Rejón http://orcid.org/0000-0002-9924-4886
Blanca Gavilán-Carrera http://orcid.org/0000-0002-9223-7181
Milkana Borges-Cosic http://orcid.org/0000-0002-8276-9387
Manuel Delgado-Fernández http://orcid.org/0000-0003-0636-9258
Fernando Estévez-López http://orcid.org/0000-0003-2960-4142

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
