## [Reviewer comments · BMJ Open]

ARTICLE DETAILS

TITLE (PROVISIONAL)	Is type of work associated with physical activity and sedentary behavior in women with fibromyalgia? A cross-sectional study from the al-Ándalus project.
AUTHORS	Girela-Rejón, Maria José; Gavilán Carrera, Blanca; Aparicio-Ortega, Esther; Borges-Cosic, Milkana; García-Rodríguez, Inmaculada C; Delgado-Fernández, Manuel; Estévez-López, Fernando

VERSION 1 – REVIEW

REVIEWER	Satoshi Kurita National Center for Geriatrics and Gerontology, Japan
REVIEW RETURNED	27-Oct-2019

GENERAL COMMENTS	This study examined the association between the type of work (productive vs. reproductive work) and the levels of physical activity and sedentary behavior in women with fibromyalgia. I have some concerns about the important part of the study. ■Introduction Although the finding is valuable to grasp physical activity of women with fibromyalgia, some studies have already reported that the type of work affects physical activity and sedentary behavior in work or non-work context. Could you state the significance of the study based on the following studies and so on? Chau JY, van der Ploeg HP, Merom D, et al. Cross-sectional associations between occupational and leisure-time sitting, physical activity and obesity in working adults. Prev Med. 2012 Mar-Apr;54(3-4):195-200. PMID: 22227284 Kurita S, Shibata A, Ishii K, et al. Patterns of objectively assessed sedentary time and physical activity among Japanese workers: a cross-sectional observational study. BMJ Open. 2019 Feb 24;9(2):e021690. PMID: 30804025 ■Methods Did the outputs from all questionnaires indicate non-normal distribution? I could not judge whether the MANCOVA was appropriate in the analysis because the premise of MANCOVA model is normal distribution.
--

REVIEWER	Anna Bergenheim Institute of Neuroscience and Physiology, Unit of Physiotherapy, Sahlgrenska Academy, University of Gothenburg, Sweden
REVIEW RETURNED	20-Dec-2019

GENERAL COMMENTS	Thank you for a well-written manuscript. I have however some concerns which I state below: Methods: p.6 The description of Total physical activity – a more complex questionnaire is mentioned, combining the LTAPI and the PAHWI. The reference needs to be given for the approach when all hours of LTPAI and PAHWI are summed up to a total score. Or, if this is the authors' own approach, it should be clearly stated. Results: I find it hard to follow all the results as given in the text. Would it be possible to present the results of the MANCOVA in tables instead? My main concern is that the severity of the main symptoms in Fibromyalgia (such as pain and fatigue) are not reported in the study. Symptom severity is normally associated with level of physical activity and work status in FM. It might be that the group who does productive work has lower symptom severity than the group who only do reproductive work, which also could be the main reason to that they are able to be engaged in productive work and more physically active. If the symptom severity of pain and fatigue is known but not presented, the manuscript would benefit from presenting this information in the table of descriptive data, and if applicable also included these variables in the analyses as covariates. If you don't have this data, this limitation needs to be addressed in the discussion. Discussion: The discussion is in some parts a bit unclear and hard to follow. Since many types of physical activity are being discussed it is important that the text is very clear to avoid misunderstanding. For example: At page 9, line 10 it says: "In particular, women with only a reproductive work did more light physical activity at home." And at page .9 line 21 it says "Our results revealed that both the group of women who did reproductive work and the group of women who also performed productive work obtained similar results in light, moderate and vigorous physical activity during their leisure time" These two sentences seem to be contradictory? Figure 3: I believe the p-value is written in a wrong way, it should be < instead of >?
---

VERSION 1 – AUTHOR RESPONSE

Reviewer 1

Comment. This study examined the association between the type of work (productive vs. reproductive work) and the levels of physical activity and sedentary behavior in women with fibromyalgia. I have some concerns about the important part of the study.

Introduction: Although the finding is valuable to grasp physical activity of women with fibromyalgia, some studies have already reported that the type of work affects physical activity and sedentary behavior in work or non-work context. Could you state the significance of the study based on the following studies and so on?

Chau JY, van der Ploeg HP, Merom D, et al. Cross-sectional associations between occupational and leisure-time sitting, physical activity and obesity in working adults. Prev Med. 2012 Mar-Apr;54(3-4):195-200. PMID: 22227284

Kurita S, Shibata A, Ishii K, et al. Patterns of objectively assessed sedentary time and physical activity among Japanese workers: a cross-sectional observational study. BMJ Open. 2019 Feb 24;9(2):e021690. PMID: 30804025

Response. Thank you for the suggestions to improve our manuscript. The above mentioned studies have been included in the discussion section (page 12). We consider these studies provide valuable evidence regarding physical activity, sedentary behavior, and their relationship with type of work depending on physical demands (mostly sitting, mostly walking, jobs etc.) in healthy adults. Our results complement these findings by examining other non-previously explored type of work categories (productive vs. reproductive) and their relation with physical activity and sedentary time in women with fibromyalgia.

Comment. Methods: Did the outputs from all questionnaires indicate non-normal distribution? I could not judge whether the MANCOVA was appropriate in the analysis because the premise of MANCOVA model is normal distribution.

Response. Normality was assumed due to large sample size. This has been specified in the methods section, page 7.

Reviewer 2

Comment. Thank you for a well-written manuscript. I have however some concerns which I state below.

Response. Thank you very much for the positive evaluation of our manuscript.

Comment. Methods: p.6 The description of Total physical activity – a more complex questionnaire is mentioned, combining the LTAPI and the PAHWI. The reference needs to be given for the approach when all hours of LTPAI and PAHWI are summed up to a total score. Or, if this is the authors' own approach, it should be clearly stated.

Response. Authors decided to combine LTPAI and PHAWI in order to obtain a total measure of physical activity. This approach has been previously used in the literature and is now stated in the methods section, page 6.

Comment. Results: I find it hard to follow all the results as given in the text. Would it be possible to present the results of the MANCOVA in tables instead? My main concern is that the severity of the main symptoms in Fibromyalgia (such as pain and fatigue) are not reported in the study. Symptom severity is normally associated with level of physical activity and work status in FM. It might be that the group who does productive work has lower symptom severity than the group who only do reproductive work, which also could be the main reason to that they are able to be engaged in productive work and more physically active. If the symptom severity of pain and fatigue is known but not presented, the manuscript would benefit from presenting this information in the table of descriptive data, and if applicable also included these variables in the analyses as covariates. If you don't have this data, this limitation needs to be addressed in the discussion.

Response. Thank you very much for raising this relevant point regarding the results format and the possible role of disease severity. Two tables with different sensitivity analyses for the comparisons between groups have been incorporated as supplementary material in order to avoid overlap with the information in the main text. These tables include unadjusted analyses, analyses accounting for original confounders and analyses including disease severity as another potential confounder. Importantly, the results were consistent with those initially presented in our work, reinforcing the strength of our

conclusions. This information has been included in the statistical analyses and results sections (pages 7 and 9).

Comment. Discussion: The discussion is in some parts a bit unclear and hard to follow. Since many types of physical activity are being discussed it is important that the text is very clear to avoid misunderstanding. For example: At page 9, line 10 it says: "In particular, women with only a reproductive work did more light physical activity at home." And at page .9 line 21 it says "Our results revealed that both the group of women who did reproductive work and the group of women who also performed productive work obtained similar resluts in light, moderate and vigorous physical activity during their leisure time" These two sentences seem to be contradictory?

Response. We agree with the reviewer in the confusion that the use of similar terms may cause. In the example given, one sentence (page 9, line 9) refers to physical activity at home, which includes house chores (cleaning, cooking, etc.) while the next sentence (page 9 line 21) refers to physical activity during leisure time. We have carefully revised the discussion to clarify any potential misleading with regard to context of physical activity or work.

Comment. Figure 3: I believe the p-value is written in a wrong way, it should be < instead of >?

Response. Thank you very much. The error has been corrected.

VERSION 2 – REVIEW

REVIEWER	Satoshi Kurita National Center for Geriatrics and Gerontology, Japan
REVIEW RETURNED	16-Feb-2020

GENERAL COMMENTS	Thank you for revising the manuscript. I considered the significance of this study is weak. Because the evidence from this study is suitable for rather specific population, the manuscript may be suitable for other journal such as Disabil Rehabil.
---

REVIEWER	Anna Bergenheim Institute of neuroscience and physiology, unit of physiotherapy, Sahlgrenska Academy, University of Gothenburg, Sweden.
REVIEW RETURNED	05-Feb-2020

GENERAL COMMENTS

Thank you for the improved and revised version of the manuscript. The authors have adjusted the manuscript according to my previous concerns and comments. I have no further comments.